# Solving Complex Manipulation Tasks with Model-Assisted Model-Free Reinforcement Learning

**Jianshu Hu**
UM-SJTU Joint Institute
Shanghai Jiao Tong University
hjs1998@sjtu.edu.cn

**Paul Weng**
UM-SJTU Joint Institute
Shanghai Jiao Tong University
paul.weng@sjtu.edu.cn

**Abstract:** In this paper, we propose a novel deep reinforcement learning approach for improving the sample efficiency of a model-free actor-critic method by using a learned model to encourage exploration. The basic idea consists in generating imaginary transitions with noisy actions, which can be used to update the critic. To counteract the model bias, we introduce a high initialization for the critic and two filters for the imaginary transitions. Finally, we evaluate our approach with the TD3 algorithm on different robotic tasks and demonstrate that it achieves a better performance with higher sample efficiency than several other model-based and model-free methods.

**Keywords:** Reinforcement learning, Data augmentation, Imaginary exploration, Optimistic initialization

## 1   Introduction

Deep reinforcement learning (DRL) has shown its potential in solving difficult robotic tasks especially when facing complex dynamics and contact-rich environment. For instance, dexterous manipulation tasks like rotating a cube or ball to a desired position and orientation can be solved with state-of-the-art model-free methods [1], model-based methods [2, 3]. However, the inherent issues related to the stability and sample efficiency of DRL algorithms still make them challenging to apply on real robots. To tackle those issues, various approaches have been investigated. In this paper, we specifically investigate how a learned model can help accelerate a model-free method. Previous research in this direction (see Section 2) used a learned model for data augmentation [4], to improve critic estimation [5], for gradient computation [6], or for guiding exploration in the true environment [7].

In contrast to previous model-based methods, we propose to "explore" inside the learned model, which is arguably safer than in the real environment. In the learned model, noisy actions are executed in states visited in the true environment to obtain imaginary transitions, which can be used to compute imaginary target Q-values for updating the Q-function of the current policy. With a perfect transition model, this approach would help accelerate learning the Q-function. However, since both the learned model and the estimated Q-function may be incorrect, directly using those imaginary transitions may lead to potential issues. To counteract them, we introduce several techniques: high Q-value initialization and filtering of imaginary transitions to favor optimistic targets that are uncertain.

Intuitively, with a higher initialization, we approximate the "optimism in face of uncertainty" principle when evaluating random actions. In the spirit of this principle, we only keep those that lead to higher evaluations than real transitions. Moreover, to avoid unnecessary updates, those imaginary transitions are selected with higher probability if there is a larger uncertainty in the corresponding Q-values. Note that since the noisy actions are performed in the learned model, our model is not a true exploration strategy. However, if such imaginary targets are indeed used to update the Q-function, these can lead to higher evaluations of the corresponding actions, which would later favor

6th Conference on Robot Learning (CoRL 2022), Auckland, New Zealand.

selecting them in the true environment. In this paper, we implement those techniques in the TD3 algorithm (see Section 3) [8]. However, they may be beneficial in other DRL algorithms as well.

**Contributions:** We propose a novel approach to exploit a learned model in DRL (see Section 4). We validate the proposed method and show that it outperforms relevant state-of-the-art algorithms in *MuJoCo* [9] robot environments and especially in some complex manipulation tasks (see Section 5) [1]. Moreover, we analyze and discuss the different proposed techniques.

## 2 Related Work

Researchers have explored various methods for combining a learned model with model-free reinforcement learning algorithms. They can mainly be divided into four categories: (1) model-based data augmentation, (2) model-based value estimation, (3) analytic gradient calculation, and (4) model-guided exploration.

Dyna [4] is a typical architecture for learning a dynamics model with the true experience and using the dynamics model to generate imaginary data for training a value function and a policy. Gu et al. [10] introduce a local linear model for representing the dynamics and show the improvement of using the imaginary rollouts with this model. ME-TRPO [11] is a method leveraging an ensemble dynamics model in a model-free reinforcement learning algorithm TRPO [12]. Another work, MA-BDDPG [13], aims to alleviate the effects of model bias by considering uncertainty when using imaginary transitions stored in a replay buffer. The uncertainty of imaginary transitions is measured as the variance of an ensemble of critics. In contrast to this method, we use the disagreement between the two target critics in TD3 [8] as an uncertainty measure and generate imaginary transitions online instead of maintaining another imaginary replay buffer.

Focusing on improving the estimation of the Q function, Feinberg et al. [5] propose a method called MVE which uses the observed states and a learned dynamics model to simulate for a fixed horizon with a current policy. With this imaginary segment, a better target Q-value (value expansion, which is a variant of n-step return) is applied in the training of the critic. Instead of using the dynamics model to forward for several steps, our method simulates only one step (to reduce the impact of the model error) but not directly with the current policy. The performance of MVE is sensitive to a difficult-to-set hyperparameter, the simulation horizon, which is limited by the quality of the learned model. To overcome this problem, Buckman et al. [14] use a weighted sum of value expansions from different horizons and different models. They learn ensemble models to approximate the transition function, reward function, and Q-function. Weights are then assigned to the value estimations of different prediction horizons according to the variance from those models. To ensure a monotonic improvement under the model bias, Janner et al. [15] provided theoretical analysis on how to decide the simulation horizon.

Instead of using the learned model as an environment simulator, researchers have also exploited its differentiability. Deisenroth and Rasmussen [16] propose a method called PILCO to learn a probabilistic dynamics model and use it in policy search by calculating analytical gradient of the object with respect to the policy parameters. Clavera et al. [6] further extend the idea and train a bootstrap ensemble probabilistic dynamics model.

As for guiding the exploration, a natural idea is to try to explore more in regions where the learned model is uncertain. For instance, Pathak et al. [17] formulate the disagreement across an ensemble model as an intrinsic reward. Another example is the work of Shyam et al. [7] in which they measure the novelty of state-action pairs with a learned model and use this novelty as the objective of an exploration Markov Decision Process (MDP) to find an exploratory policy. In contrast to these methods, we influence the exploration in the true environment by trying noisy actions in the learned model and setting a high initialization for the critics.

## 3 Background

A Markov Decision Process (MDP) is composed of a set of state $\mathcal{S}$, a set of action $\mathcal{A}$, a transition function $T : \mathcal{S} \times \mathcal{A} \rightarrow \mathcal{P}(\mathcal{S})$ (with $\mathcal{P}(\mathcal{S})$ denoting the set of probability distributions over $\mathcal{S}$), a

---

[1]Code is available at this link

reward function $r : \mathcal{S} \times \mathcal{A} \to \mathbb{R}$, and a distribution over initial state $\mu \in \mathcal{P}(\mathcal{S})$. Given a deterministic policy $\pi : \mathcal{S} \to \mathcal{A}$, the value function $V^\pi(s) = \mathbb{E}_\pi[\sum_{t=0}^\infty \gamma^t r_t \mid s_0 = s]$ is defined as the expected discounted cumulative reward an agent will receive starting from a feasible state $s$ and following the policy. The discounted factor is defined as $\gamma \in [0,1]$. To solve this MDP is to find an optimal policy $\pi^*$ that maximizes the expected value function $\pi^*(s) = \arg\max_\pi \mathbb{E}_\mu[V^\pi(s) \mid s \sim \mu]$. To find this optimal policy, we often need an action-value function $Q^\pi(s, a) = \mathbb{E}_\pi[r(s, a) + \gamma V^\pi(s')]$, which corresponds to the expected discounted sum of rewards obtained by executing action $a$ in state $s$ and acting according to policy $\pi$ thereafter. Since we focus on robotic tasks, we only consider deterministic MDPs in this work, although our approach could certainly be applied in stochastic settings as well.

**Deep Deterministic Policy Gradient (DDPG)**  DDPG [18] is a DRL algorithm with an actor-critic structure for solving an MDP with continuous state and action spaces. The policy $\pi$ (actor) and its Q-function $Q^\pi(s, a)$ (critic) are approximated by neural networks parameterized by $\theta$ and $\phi$ respectively. In DDPG, the actor interacts with the environment generating transitions that are stored in a replay buffer. At each training step, a mini-batch $\{(s_i, a_i, r_i, s_{i+1}) \mid i = 1, \cdots, N\}$ is sampled from the replay buffer and used for updating the actor's parameters $\theta$ according to the deterministic policy gradient:

$$\nabla_\theta \mathcal{L}(\pi) = \frac{1}{N} \sum_{i=1}^N \nabla_a Q(s_i, a \mid \phi) \mid_{a = \pi(s_i \mid \theta)} \nabla_\theta \pi(s_i \mid \theta), \tag{1}$$

while the critic's parameters $\phi$ are updated to minimize the following loss function:

$$\mathcal{L}(Q) = \frac{1}{N} \sum_{i=1}^N \Big( Q(s_i, a_i \mid \phi) - y_i \Big)^2 \tag{2}$$

where $y_i$ is the target Q-value for the sampled transition $(s_i, a_i, r_i, s_{i+1})$:

$$y_i = r_i + \gamma Q\Big(s_{i+1}, \pi(s_{i+1} \mid \theta') \mid \phi'\Big). \tag{3}$$

To improve the learning stability, the target Q-value is calculated with a target Q function $Q(\cdot, \cdot \mid \phi')$ and a target actor $\pi(\cdot \mid \theta')$. The target networks $(\phi', \theta')$ are initialized to the same parameters as the original networks $(\phi, \theta)$, but are then updated slowly towards the original ones with $\phi' \leftarrow (1 - \tau)\phi' + \tau\phi$ and $\theta' \leftarrow (1 - \tau)\theta' + \tau\theta$, where $\tau \in (0, 1)$ is a hyperparameter.

**Twin Delayed DDPG (TD3)**  TD3 [8] improves DDPG with three tricks: clipped double-Q learning, target policy smoothing, and delayed policy update. The first trick aims to prevent overestimation by learning two critics $Q(\cdot, \cdot \mid \phi_1)$, $Q(\cdot, \cdot \mid \phi_2)$ (with their corresponding target critics $Q(\cdot, \cdot \mid \phi_1')$, $Q(\cdot, \cdot \mid \phi_2')$). When calculating the target Q-value, the target critic with smaller value is used for training. The second trick aims to smooth the objective function by injecting a truncated Gaussian noise $\epsilon$ to the output of the target policy. Note that the perturbed actions are clipped to ensure they remain feasible. For legibility, we do not write this step in our equations. With these two tricks, the target Q-value becomes:

$$y_i = r_i + \gamma \min\{Q\big(s_{i+1}, \pi(s_{i+1} \mid \theta') + \epsilon \mid \phi_1'\big), Q\big(s_{i+1}, \pi(s_{i+1} \mid \theta') + \epsilon \mid \phi_2'\big)\}. \tag{4}$$

Lastly, the delayed policy update enforces that the actor be updated at a lower frequency than the critic. With those tricks, it has been empirically observed that the performance and stability of learning the Q-function are substantially improved.

## 4  Methodology

Our proposed algorithm (see Algorithm 1) extends TD3 to use a learned model to enhance the update of its critics. Note that while the dynamics model is not known, we assume that the reward function is known. This assumption is natural in robotics since the reward function is defined by the system designer to guide the learning of the robot.

The whole training process can be divided into two phases. In Phase 1 (time step 0 to $h$), transitions are collected by interacting with the environment using a random policy. No learning occurs up to

---
**Algorithm 1** MAMF
---
 1: initialize critics $Q(s, a \mid \phi_1), Q(s, a \mid \phi_2)$
 2: initialize actor $\pi(s \mid \theta)$, model $f$, and empty replay buffer $\mathcal{R}$
 3: set the parameters of targets $\phi_1' \leftarrow \phi_1, \phi_2' \leftarrow \phi_2$ and $\theta' \leftarrow \theta$.
 4: initialize $\eta$ and $\rho$
 5: start with initial state $s_0 \sim \mu$
 6: **for** $t = 0 \ldots h$ **do**
 7:     generate a transition with random policy and keep track of the max reward $r^*$
 8:     save transition $(s_t, a_t, r_t, s_{t+1})$ in replay buffer $\mathcal{R}$
 9:     **if** episode ends **then** reset the environment $s_{t+1} \sim \mu$ **end if**
10:     **if** $t > h_0$ **then** update $f$ with $\{(s_i, a_i, r_i, s_{i+1}) | i = 1, \cdots, N\}$ sampled from $\mathcal{R}$ **end if**
11: **end for**
12: use max reward $r^*$ to set the bias in the critics according to Equation (6).
13: start with initial state $s_0 \sim \mu$
14: **for** $t = h+1, \ldots, H$ **do**
15:     interact with the environment with current policy $\pi$ and small noise
16:     save transition $(s_t, a_t, r_t, s_{t+1})$ to replay buffer $\mathcal{R}$
17:     **if** episode ends **then** reset the environment $s_{t+1} \sim \mu$ **end if**
18:     sample $\{(s_j, a_j, r_j, s_{j+1}) \mid j = 1, \cdots, N\}$ from $\mathcal{R}$ and train $f$
19:     sample $\{(s_k, a_k, r_k, s_{k+1}) \mid k = 1, \cdots, N\}$ from $\mathcal{R}$ and create imaginary transitions
20:     $\eta \leftarrow \eta \times \rho$
21:     filter the imaginary transitions with the two filters
22:     update critic according to Equation (10)
23:     **if** $t \bmod policy\_delay = 0$ **then**
24:         update actor
25:         update target networks
26:     **end if**
27: **end for**
---

time step $h_0$ after which the dynamics model starts to be trained. Data from Phase 1 is also used to set a high initialization for the critics. During Phase 2 (time step $h + 1$ to $H$), transitions are now generated by the actor and both actor and critic start to be trained. The dynamics model is further updated online and used for creating imaginary transitions with noisy actions.

The goal in our approach is to train the critics with those imaginary transitions, which can either improve the Q-estimation or guide the exploration such that potentially promising actions could be tried in the true environment. To that aim, we apply two filters to the imaginary transitions, since using any imaginary transitions is generally detrimental due to model bias and Q-value estimation error. The first filter keeps the imaginary transitions with higher target Q-values. The second filter selects imaginary transitions whose Q-values are uncertain with higher probability. With all these components, we propose the model-assisted model-free (MAMF) algorithm (see Algorithm 1). To ensure the reproducibility of our results, the source code of MAMF will be released after publication.

Next we explain how we learn the dynamics model, how we implement the high initialization, how we perform exploration in the learned model, and how we filter the imaginary transitions.

**Learning the Dynamics Model.** Our algorithm starts by training a dynamics model $\mathcal{M} : \mathcal{S} \times \mathcal{A} \to \mathcal{S}$ with samples generated by a random policy (e.g., which uniformly randomly selects actions). Before starting updating $\mathcal{M}$, $h_0$ transitions are first collected. Then, $\mathcal{M}$ is repeatedly updated via stochastic mini-batch gradient descent until $h$ transitions have been generated by the random policy, which ends Phase 1. After that, the model is still updated with the same procedure, but the transitions are now generated by the current policy. Online DRL training is performed until $H$ transitions have been generated. The model is (pre)trained by minimizing a mean-squared error between predicted next states and true next states computed over a mini-batch $\{(s_i, a_i, r_i, s_{i+1}) \mid i = 1, \cdots, N\}$:

$$L_{\mathcal{M}} = \frac{1}{N} \sum_{i=1}^{N} \left( f(s_i, a_i) - s_{i+1} \right)^2. \tag{5}$$

**High Initialization of Critics.** The critics are initialized to optimistic values by changing the bias term in the last layer of the critic networks. This bias is set as follows:

$$b = (r^* \times l) \times c \tag{6}$$

where $r^*$ is the max reward observed during Phase 1, $l$ is the max episode length, and $c \in [0, 1]$ is a hyperparameter controlling how optimistic the high initialization is.

**Exploration with Learned Model.** In the actor-critic architecture of TD3, the critic is trained using transitions $\{(s_i, a_i, r_i, s_{i+1}) \mid i = 1, \cdots, N\}$ sampled from the replay buffer. For the same set of states $\{s_i \mid i = 1, \cdots, N\}$, a noise from a truncated normal distribution $\nu \sim \text{clip}\big(\mathcal{N}(0, 1), -\eta, \eta\big)$ with a large clipping bound $\eta$ is applied on the policy action $\pi(s_i \mid \theta)$ to generate imaginary transitions. The bound $\eta$ is initialized as the max action $a_{\max}$ the agent can take in each dimension. After each iteration, the bound for truncating the Gaussian noise exponentially decays with a fixed decaying rate $\rho$. Formally, the imaginary transitions $(s_i, \hat{a}_i, \hat{r}_i, \hat{s}_{i+1})$ generated by the learned model $f$ and known reward function $r$ are written:

$$\hat{a}_i = \text{clip}\big(\pi(s_i \mid \theta) + \nu, -a_{\max}, a_{\max}\big), \qquad \hat{s}_{i+1} = f(s_i, \hat{a}_i), \qquad \hat{r}_i = r(s_i, \hat{a}_i). \tag{7}$$

**Filtering.** To only use the imaginary transitions whose actions are potentially better than the original ones, these transitions are filtered by their target Q-values. Only the imaginary transitions whose target Q-values are higher than the original ones are considered:

$$\hat{y}_i(\hat{r}_i, \hat{s}_{i+1} \mid \theta', \phi_1', \phi_2') \geq y_i(r_i, s_{i+1} \mid \theta', \phi_1', \phi_2') \tag{8}$$

where $\hat{y}_i$ and $y_i$ are calculated by Equation (4)

The remaining imaginary transitions are filtered with respect to the uncertainty in the Q-value estimates. This uncertainty can be measured as the disagreement of the two target critics in TD3, which can be expressed as the absolute value of the difference of the target Q-values given by the two target critic networks:

$$\Delta = |Q(\hat{s}_{i+1}, \pi(\hat{s}_{i+1} + \epsilon \mid \theta') \mid \phi_1') - Q(\hat{s}_{i+1}, \pi(\hat{s}_{i+1} + \epsilon \mid \theta') \mid \phi_2')|. \tag{9}$$

This difference is then normalized by the max difference observed so far in the sampled transitions to obtain a ratio $P = \frac{\Delta}{\max \Delta} \in [0, 1]$. With probability $P$, an imaginary transition is selected for training the critics, otherwise it is dropped. Therefore, imaginary transitions with larger uncertainty have a higher chance to be kept. The number of remaining imaginary transitions after the two filters is denoted by $n$.

Finally, the loss for the critics is composed of two parts, the original loss and an additional loss obtained from the filtered imaginary transitions:

$$L_\phi = \frac{1}{N} \sum_{i=1}^{N} \big(Q(s_i, a_i \mid \phi) - y_i\big)^2 + \frac{1}{n} \sum_{i=1}^{n} \big(Q(s_i, \hat{a}_i \mid \phi) - \hat{y}_i\big)^2 \tag{10}$$

where the target value $y_i$ is calculated with Equation (4) and the target values $\hat{y}_i$ for imaginary transitions follow almost the same calculation but with the action, next state, and reward substituted by their imaginary counterparts.

## 5 Experimental Results

Our experiments are designed to answer the following questions:

- Do our propositions improve the performance with respect to relevant baselines?
- Does each component of our method (i.e., decaying noise, high initialization, filtering) contribute to its performance?
- In which aspects does our method help?

Before answering those questions, we present next the experimental environments and discuss the experimental set-up we used.

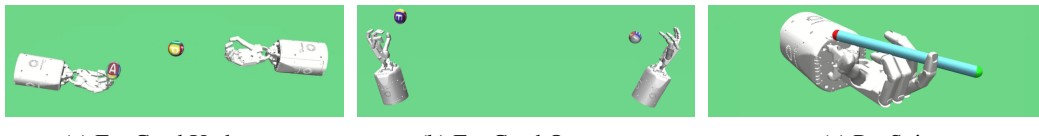


(a) EggCatchUnderarm      (b) EggCatchOverarm      (c) PenSpin

Figure 1: Visualization of three environments in Dexterous Gym


## 5.1 Environments and Experimental Set-Up

We evaluated our approach on several robot environments in *MuJoCo* [9] and three complex manipulation tasks proposed in Dexterous Gym [19]. The visualizations of these manipulation tasks are shown in Figure 1. The task is throwing and catching an object at desired position and orientation with two dexterous manipulation hands in EggCatchUnderarm and EggCatchOverarm. The task of PenSpin is rotating a pen without dropping it. The details of these environments are described in Appendix A. We chose these three representative environments from Dexterous Gym [19]. The other environments like BlockCatchUnderarm are simply variants with different objects.

Across all the environments, all the hyperparameters such as learning rates, batch size and network architectures for the actor and critic are kept the same except for the hyperparameters related to setting the bias. The hyperparameters and a brief discussion about setting them are shown in Appendix B. All the experiments are performed over five different random seeds and the performance is measured by the sum of rewards averaged over 10 episodes.

## 5.2 Results

### 5.3 Do our propositions improve the performance w.r.t relevant baselines?

To answer this question, we evaluate our method on the environments introduced above. We compare with the following baselines: (1) TD3, (2) MVE-TD3: MVE with TD3, and (3) MA-TD3: MA-BDDPG with TD3. In Appendix C, we present some additional experiments where we compare and integrate our approach with the state-of-the-art model-free RL method called SAC [20], which encourages exploration by incorporating an entropy term.

Both MVE [5] and MA-BDDPG [13] are methods that use a dynamics model to generate imaginary transitions for training the critics. Since their source codes are not publicly available, we implemented those methods in TD3 for consistency. For MVE, we set the prediction horizon to 3 which is applicable in all the environments. For MA-BDDPG, we actually implemented a version similar to it, which corresponds to TD3 with only imaginary transitions filtered by uncertainty. It is equivalent to our method without high initialization, decaying noise and the filter favoring higher target Q values. Similar to the original method proposed by Charlesworth and Montana [19], demonstrations are also used in all the experiments in Dexterous Gym.

The training results are shown in Figure 2. We can see the outperformance of our method across different environments. Especially in complex environments like EggCatchUnderarm, our method achieves a much higher final return and has low variability across different runs. We also notice that combining MVE with TD3 can not guarantee an improvement in some environments. This might be due to the target Q smoothing in TD3 (clipped noise added to the action when calculating the target Q-values). Similar results of MVE can also be observed in the work by Buckman et al. [14]. Their results also show that MVE does not guarantee an improvement when applied on DDPG [18].

### 5.4 Does each component of our method contribute to its performance?

To prove the significance of each component in our method, we performed an ablation study. By comparing the results shown in Figure 3(a)-(b), we can find that high initialization is necessary to counteract the effect of the model bias and encourage the exploration at the beginning. Using a decaying noise and choosing imaginary transitions with higher target Q-values achieve the ideas of both seeking optimistic state-action during training and of exploring in the model. The filter with uncertainty guides the learning of the Q-function to emphasize on the uncertain part and thus helps the estimation of Q-values. All the components in synergy improve the training of the critic.

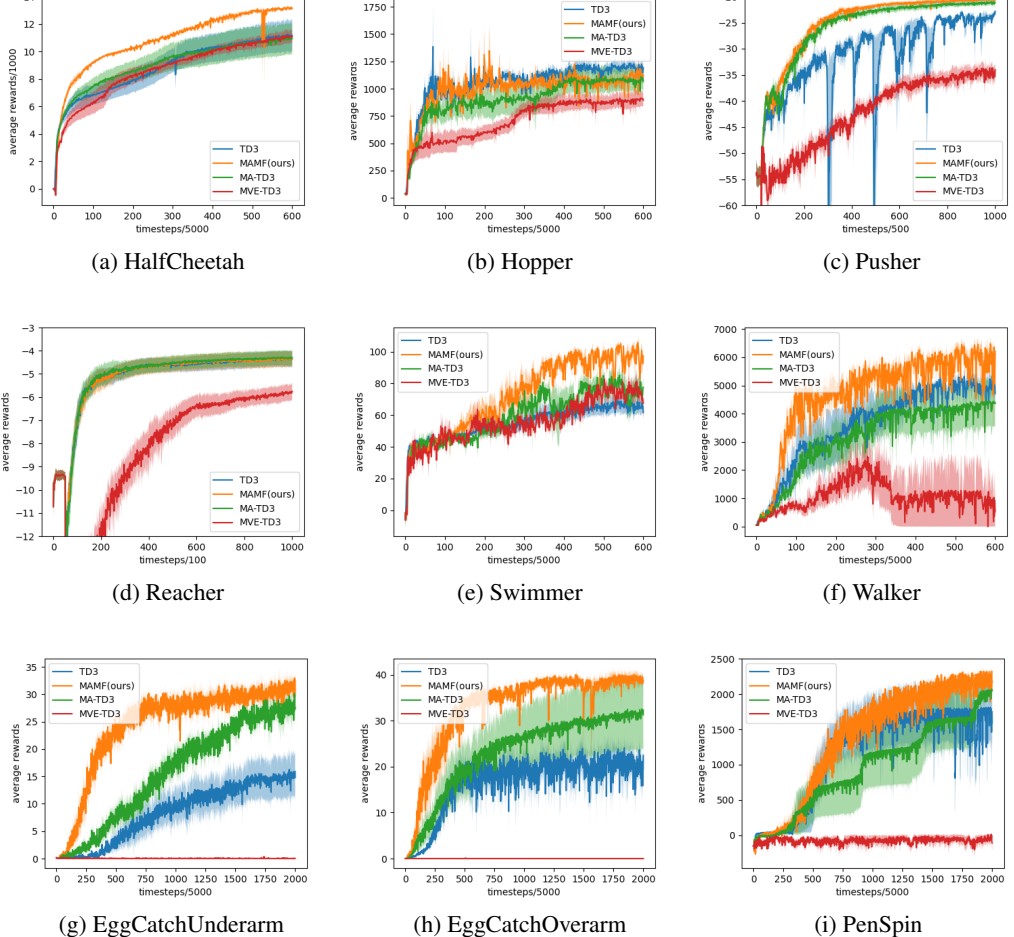

Figure 2: Results of training in *MuJoCo* environments and Dexterous Gym

## 5.5 In which aspects does our method help?

We conjecture that our method helps in three ways: better exploration, better estimation of Q-values, and exploitation of independence among action components. For the first point, we believe that the higher final performance of our method compared to the baselines provides some evidence of the improved exploration.

For the second point, we compare our method with a variant of TD3 where critics are updated more frequently. This variant could achieve better Q-estimations and thus better performances in some environments. Due to the page limit, the results and details for the experiments are shown in Appendix C. We can see that simply updating the critic more often is still not sufficient to achieve the performance gains of our method.

For the last point, we believe that our method also exploits potential action independence (i.e., in some states, parts of the actions do not have an impact on the next environment state). This kind of independence is common in robotic tasks e.g., in a walking robot, the actions of any limb that does not touch the ground may have less effects on the next states, or more concretely, in EggCatchUnderarm, once a hand has thrown a ball, the actions of this hand is not important anymore. To test this conjecture, we augment HalfCheetah by doubling its action space $\mathcal{A}$ to an enlarged action space $\mathcal{A} \times \mathcal{A}$. The transitions in this augmented environment depend only on the first or second part of the enlarged actions. This dependence switches from one part to the other every $H$ time steps. Following this pattern, part of the actions has no direct effect. The results of training in this arti-

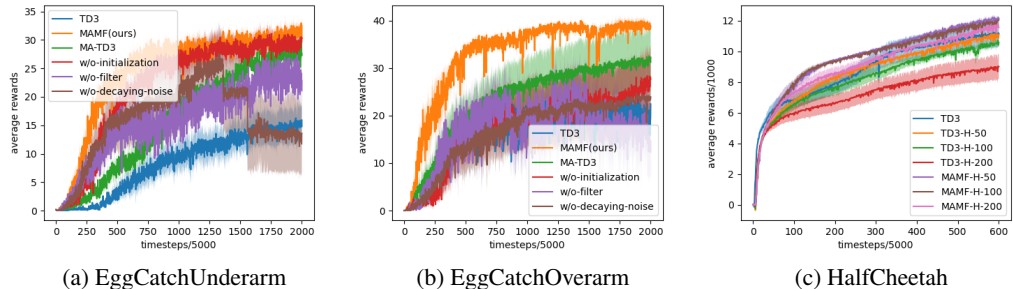

|  (a) EggCatchUnderarm | (b) EggCatchOverarm | (c) HalfCheetah |

Figure 3: Figures (a) and (b) show the ablation study. Figure (c) show the results of training in the environments defined in Section 5.5

ficial environment with different frequencies of switching are shown in Figure 3(c). Although the augmented environment has a larger action space, our method can still improve the performance.

# 6 Limitations

The limitations of our work mainly include these three aspects:

(1) Using a too large or too small bias in the critic will hurt the performance of our method. Although our heuristic for setting the bias is quite robust (except for HalfCheetah), it is not clear how to analytically determine it. A more theoretical analysis may be required for applying it in a more general situation.

(2) While much work has investigated exploration strategies in model-free algorithms, in our method, a simple truncated Gaussian noise is applied to generate imaginary transitions. A more sophisticated way of choosing exploratory actions might further improve our approach.

(3)Although our proposed method significantly improves the sample efficiency and the performance, especially on the harder tasks of Dexterous Gym, we believe that more efforts may still be needed to fully run the algorithm in a real robot. Potential avenues to make the approach even more practical could be to use it in combination with sim2real methods [21, 22], exploit any extra a priori known information such as symmetries [23], or exploit the learned model in some other ways [24] for instance.

# 7 Conclusion

We propose a method called MAMF, which leverages a dynamics model to help train the critic in a model-free algorithm. One key novelty is to generate imaginary data with exploratory actions. Our experiments demonstrate the sample efficiency and performance improvements of this method even in some complex manipulation tasks. We believe that our method is a step forward towards making DRL practical on real robots and shows a promising way of solving those complex tasks. Although our proposition is implemented with TD3, it is actually independent from this DRL algorithm. Thus, one interesting future work would be to adapt and combine our proposition with other model-free algorithms.

**Acknowledgments**

This work is supported in part by the program of National Natural Science Foundation of China (No. 62176154) and the program of the Shanghai NSF (No. 19ZR1426700).

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
