# OpenReview forum: "Solving Complex Manipulation Tasks with Model-Assisted Model-Free Reinforcement Learning"
_robot-learning.org/CoRL/2022/Conference — CoRL 2022 Poster_

### Official Review · Reviewer_4nCH · 2022-07-25

**Originality:** Fair
**Technical Quality:** Good
**Clarity Of Presentation:** Very Good
**Impact:** 3

**Recommendation:**

Weak Reject: I recommend rejecting the paper, but will not argue for my recommendation if the majority of other reviewers have a different opinion.

**Summary:**

This paper presents a deep reinforcement learning approach to improve the sample efficiency of an actor-critic method. It adds random noise to action values to create imaginary transitions. To address the model bias from using the transitions, it introduces high Q-value initialization of critics and two filtering methods that favor transitions with high target Q-values and high uncertainty in the Q-value estimates. They compare their method with different variants of TD3.

**Issues:**

Results:
* [5.2.1] It will be clearer if there is an additional sentence that describes the details of MVE, similar to what you did for MA-BDDPG.

Conclusion:
* The sentence “one key novelty is to generate imaginary data with exploratory actions” is not true. Some prior works also add Gaussian noise to the action values, such as [1].

Typos:
* Line 46: “see sec:result”.
* Line 179: there is an extra “to”.

**References:**

[1] Hado van Hasselt and Marco A. Wiering. Reinforcement Learning in Continuous Action Spaces, 2007.


**Quality Of The Limitations Section:**

Limitations are addressed clearly

**Reviewer Expertise:**

4: The reviewer is confident but not absolutely certain that the evaluation is correct

**Robotics Focus:**

Highly relevant to robotics but no hardware experiments

**Strengths And Weaknesses:**

**Strengths:**
* This paper presents several interesting tricks to counteract the model bias of using imaginary transitions.
* The paper is well structured and written.
* Figures are clear and useful for understanding the results.

**Weaknesses:**
* Since the proposed method pre-trains a dynamics model, it is unfair not to show the pre-training steps in the experimental results (Figures 2 and 3) when compared to the baselines. The proposed method's sample efficiency could be overestimated as the number of pre-training steps is not taken into account.
* The experimental results in Figure 2 will look more convincing if non-TD3 algorithms are used for comparison, such as SAC. Figure 2 looks more like an ablation than a baselines comparison since TD3 and MA-TD3 are implementations without some of the components in your method.
* Lack of real robot experiments.


**Summary Of Recommendation:**

This paper presents an interesting method to improve the sample efficiency of an actor-critic method. However, two issues need to be addressed. First, the proposed method pre-trains a dynamics model in the environment, but the number of pre-training steps is not shown in the experimental results, resulting in an unfair comparison. Second, most baselines (TD3 and MA-TD3) in Figure 2 are implementations without some of the proposed method's components, making it more like an ablation study than a baselines comparison. The authors should consider adding state-of-the-art non-TD3 methods in Figure 2. Additionally, having real robot experiments will certainly improve the paper.

---

> ### Author Response · Authors · 2022-08-26
> **Response to Reviewer 4nCH**
>
> **Comment:**
>
> Thank you for showing interest in our method and noting the clarity of our presentation. Below, we answer to the issues that were not addressed yet by our response to the meta-reviewer.
>
> Weaknesses:
>
> * W1: The number of pre-training steps is not shown in the experimental results, resulting in an unfair comparison.
>
> Please see our answer to the meta-reviewer about pretraining in W3.
>
> * W2: The authors should consider adding state-of-the-art non-TD3 methods in Figure 2.
>
> We ran SAC on all the tasks we considered. TD3-MAMF is generally better than SAC. Please also refer to our response to the meta-reviewer about SAC in W5.
>
> * W3: Having real robot experiments will certainly improve the paper.
>
> Please see our answer to the meta-reviewer about real-robot experiments in W2.
>
> Issues:
>
> * I1:  [5.2.1] It will be clearer if there is an additional sentence that describes the details of MVE, similar to what you did for MA-BDDPG.
>
> We have updated our previous description of MVE in lines 64-69 by mentioning the form of their value expansion. Hopefully, this makes the details of MVE clearer.
>
> * I2: The sentence “one key novelty is to generate imaginary data with exploratory actions” is not true. Some prior works also add Gaussian noise to the action values, such as [1].
>
> In [1], noisy actions are performed in the RL environment. In our work,noisy actions are performed in the learned model, not the RL environment. This is novel to the best of our knowledge. Using the learned model to try exploratory actions may arguably be safer.
>
> * I3: Typos.
>
> Thanks for indicating the typos. We have updated the paper.
>
>
>
> **Zip File:**
>
> /attachment/6125ab7cb86dd9865f419f6e43c74aa121996494.zip

---

### Official Review · Reviewer_4eEP · 2022-07-26

**Originality:** Good
**Technical Quality:** Good
**Clarity Of Presentation:** Good
**Impact:** 3

**Recommendation:**

Weak Accept: I recommend accepting the paper, but will not argue for my recommendation if the majority of other reviewers have a different opinion.

**Summary:**

This paper studies improving the sample efficiency of model-free DRLs using a learning transition model. The proposed method uses a learning model to generate imaginary transitions, which are then used to update the critic. The authors applied the proposed method to various simulated robot tasks and demonstrated that it outperformed other methods with high sample efficiency.

**Issues:**

It would be necessary to explain how the proposed method can be used to train real robot tasks by comparing it with previous sample-efficient methods.

**Quality Of The Limitations Section:**

Additional details required

**Reviewer Expertise:**

3: The reviewer is fairly confident that the evaluation is correct

**Robotics Focus:**

Relevant but unlikely to deploy to hardware in near future

**Strengths And Weaknesses:**

Strengths:
* When the proposed method is applied to TD3, sample efficiency and performance are improved simultaneously
* Comparison with other methods on various robot simulation tasks
* An ablation study of the proposed method is conducted

Weaknesses:
* The proposed method is a general method to improve DRL sample efficiency. It is not specialized for the complex manipulation tasks in the title. The association between the proposed method and complex manipulation is not strong.
* No comparison in real robot task
* As the authors describe in their limitations, the sample efficiency is not high enough to learn complex manipulations on a real robot. The combination of sim2real and the proposed method is given as an example as a solution; however, the proposed method would not be necessary if a simulator could be used.
* No comparison of applying the proposed method to Soft Actor Critic (SAC), which is more sample efficient than TD3.


**Summary Of Recommendation:**

The approach is general enough to apply to a variety of tasks. Simulation results show high sample efficiency and performance for various robot tasks.

---

> ### Author Response · Authors · 2022-08-26
> **Response to Reviewer 4eEP**
>
> **Comment:**
>
> Thank you for noting the improved performance of our method and the extent of our experimental evaluation. Below, we answer to the issues that were not addressed yet by our response to the meta-reviewer.
>
> * W1: The proposed method is a general method to improve DRL sample efficiency. It is not specialized for the complex manipulation tasks in the title. The association between the proposed method and complex manipulation is not strong.
>
> Although our method is generic, we believe that it really shows its potential on complex robotic manipulation tasks such as EggCatchUnderArm or EggCatchOverArm, where TD3-MAMF greatly outperforms TD3.
>
> * W2: No comparison in real robot task.
>
> Please check our answer to meta-reviewer in W2
>
> * W3: As the authors describe in their limitations, the sample efficiency is not high enough to learn complex manipulations on a real robot. The combination of sim2real and the proposed method is given as an example as a solution; however, the proposed method would not be necessary if a simulator could be used.
>
> We believe that accelerating RL training even in simulation is still desirable. It allows to iterate faster and perform more experiments if needed. However, more importantly, note that our method does not only accelerate learning, it also allows to obtain better policies, e.g., Half-Cheetah, Swimmer, or the dexterous gym tasks.
>
> * W4: No comparison of applying the proposed method to Soft Actor Critic (SAC), which is more sample efficient than TD3.
>
> Please see our answer to meta-reviewer in W5.
>
> **Zip File:**
>
> /attachment/2966b096f7c4d3cba02eebec4998d7a35de87ff9.zip

---

### Official Review · Reviewer_jm78 · 2022-07-30

**Originality:** Good
**Technical Quality:** Good
**Clarity Of Presentation:** Very Good
**Impact:** 4

**Recommendation:**

Weak Accept: I recommend accepting the paper, but will not argue for my recommendation if the majority of other reviewers have a different opinion.

**Summary:**

The paper presents an algorithm that combines model based and model free RL for control. Specifically the authors propose a learning framework where a learned model is used, after being pre-trained, to roll-out imaginary transitions that are used to update a critic in order to eventually facilitate exploration in the real environment to solve a task. The authors present experimental evaluation of the proposed algorithms on a variety of simulated environments and compares to a number of baselines.

**Issues:**

see above

**Quality Of The Limitations Section:**

Limitations are addressed clearly

**Reviewer Expertise:**

4: The reviewer is confident but not absolutely certain that the evaluation is correct

**Robotics Focus:**

Highly relevant to robotics but no hardware experiments

**Strengths And Weaknesses:**

Strength: The paper is written clearly and is easy to follow, the algorithm is also presented well and it would be in principle possible to reimplement it. The direction of combining model based and model free is relevant and interesting and looking at the model as a possible avenue for doing exploration is also an interesting direction. The results overall seem promising and show that the method works.

Weaknesses: I am not super convinced by the results, compared to TD3 there might not be a clear benefit as the model was also already pre-trained. The final performance is mostly comparable to TD3. I would not be so worried about this itself, however since you don't show any transfer/generalisation experiments learning a model without reusing it, where TD3 actually also performs well, seems a bit of an overhead to me.

I think the paper would be very strong if you could show that you can reuse the model across tasks, or are you doing this already but did not make clear in the paper?

I think that for that it would also be important to clarify the pre-training of the model more, and if you pre-trained for each experiment a new model.

- robot experiments are missing

Questions:

- could you give a better intuition or understanding of why the initialisation of the bias help? Also how did you come up with this equation, what why were you expecting this would help?

- How much data did you use to pre-train and how close is this data to the experimental domain?

- Why did you not compare to a purely model based RL algorithm like this one https://arxiv.org/pdf/1912.01603.pdf ?

**Summary Of Recommendation:**

There are a couple of questions that need clarification but I'm willing to adjust my assessment during the authors respond period.

---

> ### Author Response · Authors · 2022-08-26
> **Response to Reviewer jm78**
>
> **Comment:**
>
> Thank you for noting the quality of our exposition and describing our approachas interesting and promising. Below, we provide a rebuttal to the main raised issues and answer the questions next. Please also refer to our response to the meta-reviewer
>
> Weaknesses:
>
> * W1: I think that for that it would also be important to clarify the pre-training of the model more, and if you pre-trained for each experiment a new model.
>
> Please see our answer to the meta-reviewer about pretraining in W3.
>
> The initial phase when only the dynamics model is trained occurs for each control task.
>
> * W2: I am not super convinced by the results, compared to TD3 there might not be a clear benefit as the model was also already pre-trained. The final performance is mostly comparable to TD3.
>
> First of all, we would like to emphasize that our method does not benefit from any additional data or training steps compared to the other methods. We believe that Figure 2 clearly shows that in many tasks (e.g.,Half-Cheetah, Swimmer, Dexterous gym tasks) the final performance ofthe policy learned by our method greatly outperforms the standard TD3
>
> * W3: I would not be so worried about this itself, however since you don't show any transfer/generalisation experiments learning a model without reusing it, where TD3 actually also performs well, seems a bit of an overhead to me.
>
> While learning a dynamics model is indeed an overhead, the advantageof our proposition for TD3 is that learning is accelerated and the final performance is never lower and often much greater compared to baselines. We believe that decreasing sample complexity by increasing computational costs is a good trade-off for robotics.
>
> Please check our answer to the meta-reviewer about transferring in W4.
>
> * W4: Robot experiments are missing
>
> Please see our answer to the meta-reviewer about real-robot experiments in W2.
>
> Questions:
>
> * Q1: Could you give a better intuition or understanding of why the initialisation of the bias help? Also how did you come up with this equation, what why were you expecting this would help?
>
> The high initialization leads to an approximation of the principle of ”optimism in the face of uncertainty”. Actions that haven’t been tried or have only been tried a few times will generally have a higher Q-value thanks to this high initialization. This will indirectly promote the choice of those actions by the actor, which influences the exploration in the real environment.
>
> A good value for this high initialization is domain-dependent and needs to be larger than the Q(s, a) of the initial policy for any state and action. One natural upper-bound is given by the best reward observed during the initial phase multiplied by the length of an episode. Since this value is too optimistic and may not even be achievable, we define our proposition as a controllable percentage of that upper-bound. We believe this heuristic formula is simple and easy to understand.
>
> * Q2: How much data did you use to pre-train and how close is this data to the experimental domain?
>
> Our whole training procedure is divided into two phases. During Phase1, only the dynamics model is trained and during Phase 2, the dynamics model and both actor and critic are updated. The data used for training the dynamics model always comes from the RL environment, but in Phase1, it is generated by a random policy, while in Phase 2, it is generated by the actor.
>
> During Phase 1, the amount of used data is upper-bounded by the duration of Phase 1. For a simple task like Reacher, it is 5000 time steps, but for more complex tasks, it can be up to 25000 time steps. This information is provided in Table 2 in the appendix of the updated paper.
>
> * Q3: Why did you not compare to a purely model based RL algorithm like this one https://arxiv.org/pdf/1912.01603.pdf?
>
> The suggested reference is a model-based RL approach that aims at learning a dynamics model in latent space when the RL agent receives high-dimensional observations. In our work, we assume that the state information is available and our goal is to accelerate RL training for complex control tasks. Therefore, we do not need to learn a dynamics model from high-dimensional observations. Extending our approach to that setting would certainly be interesting. We leave this to future work.
>
>
>
>
> **Zip File:**
>
> /attachment/43ab748e185b725e9ddcc09e65f8d819ede9dd6c.zip

---

### Official Review · Reviewer_BZ4p · 2022-07-31

**Originality:** Good
**Technical Quality:** Very Good
**Clarity Of Presentation:** Very Good
**Impact:** 3

**Recommendation:**

Weak Accept: I recommend accepting the paper, but will not argue for my recommendation if the majority of other reviewers have a different opinion.

**Summary:**

This paper proposes an approach (called MAMF) to augment a model-free reinforcement learning algorithm (namely TD3) with a learned dynamics model. The learned model is used to sample imaginary transitions from actually observed states, which are then used to train the critic. The imaginary transitions are filtered based on two criteria: 1) higher Q values, and 2) higher uncertainty. Experimental results in simulation show that this method outperforms prior methods in simulated manipulation tasks, achieving higher performance and better sample efficiency.


**Issues:**

I couldn't find/think of anything that needed revision.

**Quality Of The Limitations Section:**

Limitations are addressed clearly

**Reviewer Expertise:**

4: The reviewer is confident but not absolutely certain that the evaluation is correct

**Robotics Focus:**

Highly relevant to robotics but no hardware experiments

**Strengths And Weaknesses:**

Strengths:

The paper is clear and easy to read, and has the relevant references (to the best of my knowledge). It also clarifies the differences with other closely related prior approaches well. Experiments are performed on many diverse environments, showing the effectiveness of the approach across domains and problem types. Ablation studies are also presented for various aspects of the algorithm, which is useful to see. I believe a key part of what makes this method works well is that the learned model is only used for extrapolating a single time-step into the future, limiting the amount of bias introduced by an inaccurate model. Another good feature is that the proposed strategy is not necessarily limited to augmenting TD3: it could (in theory) be applied to any model-free RL algorithm, although this is not tested in the paper. This definitely extends the applicability of the algorithm.

Weaknesses:

I believe the main weakness of this paper is probably the lack of real-world robot experiments. However, I believe this is mitigated by: 1) extensive experiments in a variety of simulation domains, and 2) the generality of the approach to any model-free RL algorithm, not just TD3.

Another potential issue which could make real-world application more challenging is the difficulty of training dynamics models for high-dimensional state/observation spaces such as camera images.


**Summary Of Recommendation:**

This is a good paper, which experimentally evaluates a model-based augmentation to model-free RL algorithms. While there are some weaknesses and limitations when considering real-world application, it nonetheless seems like a useful strategy to consider, and I would consider this a good contribution to the RL community.

---

> ### Author Response · Authors · 2022-08-26
> **Response to Reviewer BZ4p**
>
> Thank you for noting the extensiveness of our experiments and the generality of our proposition. Below, we answer to the weaknesses mentioned in the review.
>
> * W1:  Lack of real-world robot experiments
>
> Although we have not conducted any experiments with real robots yet, we believe that our approach can be seen as a step towards making deepRL-learned policy more practical on real robots. By learning faster and obtaining better policies, experiments on real robots can be accelerated and facilitated by transferring from those learned policies
>
> * W2: The difficulty of training dynamics models for high-dimensional state/observation spaces
>
> We agree that extending our approach to RL with high-dimensional inputs would be interesting. Some recent proposed methods, such as Dreamer as suggested by Reviewer jm78, could possibly be combined with our proposition. We leave this extension to future work.

---

### Author Response · Authors · 2022-08-26
**Revised Manuscript**

**Comment:**

Thank you for reading our paper and providing valuable feedbacks!

We have updated our paper and uploaded new material for addressing the issues.

**Zip File:**

/attachment/c69de0c3eac377056e0752818d9b8e91d018060b.zip

---

### Meta-Review · Area_Chair_FrK3 · 2022-08-02

**Recommendation:** Accept (Poster)
**Confidence:** 4

**Metareview:**

Strengths:
+ Reviewers agree that the method proposed is technically sound and well explained.
+ An extensive set of experiments in simulation seems to indicate a sample efficiency and performance advantage over the TD3 baseline.
+ The method is general (applicable to more than manipulation tasks, and applicable on top of more algorithms than TD3)

Weaknesses
- There are no experiments on real robots.
- The pre-training time is not considered in the comparisons, which seems unfair if the main point is improved sample efficiency. The authors have addressed and clarified this aspect in discussion.
- Re-using the learned model on a different task would highlight the advantages of this method, but that aspect is not tackled. The authors did attempt to investigate this aspect in the discussion period, but with limited results. This remains a key area for improvement for future studies.
- Showing performance improvements on top of a different algorithm than TD3 (e.g. SAC) would improve the paper, but that aspect is not tackled. Again, the authors attempted to address this, but in the limited time available during discussion the results were mixed. This also remains a key area for improvement for future studies.

**Best Paper Nomination:**

No

---

> ### Author Response · Authors · 2022-08-26
> **Response to Area Chair FrK3**
>
> **Comment:**
>
> Thank you for summarizing the strengths and weaknesses mentioned by the reviewers. We would also like to thank all the reviewers for their constructive suggestions. Below, we clarify some misunderstandings (performance, pretrain-ing) and provide an answer to the weaknesses:
>
> * W1: The improvements over TD3 seem limited ...
>
> We are not sure we understand this point. In the Mujoco tasks, our method generally outperforms TD3, often to a large extent. The only exceptions are Hopper and Reacher where the performances are relatively similar. On the more complex tasks (Dexterous gym), our method both learns faster and reaches a higher final performance than TD3.
>
> * W2: There are no experiments on real robots.
>
> Although we have not conducted any experiments with real robots yet, we believe that our approach can be seen as a step towards making deepRL-learned policy more practical on real robots. By learning faster and obtaining better policies, experiments on real robots can be accelerated and facilitated by transferring from those learned policies.
>
> * W3: The pre-training time is not considered in the comparisons ...
>
> We are sorry that we were not careful enough in our description of our method. What we called pretraining is simply the initial phase of our whole training process. This initial phase is naturally counted as part of the training for our method. Therefore, the experimental comparison with the baselines is completely fair. Our method is not advantaged with any additional training data.
>
> To avoid any misunderstanding about this point, we do not refer to this initial phase as pretraining anymore. We have updated our paper (see lines 134-140 and Alg.1).
>
> * W4: Re-using the learned model on a different task ...
>
> First of all, note that our main goal with the learned dynamics model is to demonstrate that some form of exploration with it can be beneficial, which is novel to the best of our knowledge. Our method includes an initial phase where only the dynamics model is trained to ensure a suﬀicient quality of this model before it is used. If an already-learned model is transferred to a new task (in an environment with the same dynamics), we conjecture that it can help learn this new task faster.
>
> Note that in our current experiments, all the tasks have different state/action spaces, which prevent any direct transfer. To prove the conjecture, we have performed some new experiments in Half-Cheetah and EggCatchUnderarm (where we change the reward function to obtain a new task). We learn a dynamics model for the new task and use this learned model to initialize the dynamics model for the original task. The results are shown in the attached figures. It takes us four days for evaluating one algorithm in Dexterous gym, so the experiments in EggCatchUnderarm are not finished yet due to the time constraint. But we will continue to run it and report the results in the final version.
>
> From current results, the performance of transferring a learned model is better in EggCatchUnderarm while it is slightly worse in Half-Cheetah. We think the learned model is trained to be good around the current policy and may not be so helpful for other policies. And a randomly-initialized model may also implicitly help with exploration.
>
> * W5: Showing performance improvements on top of a different algorith than TD3 ...
>
> Note that we originally focus on TD3 because we believe that deterministic policies may be preferred in robotics. However, we agree that evaluating our proposition with SAC would further demonstrate its benefit. Therefore, we ran SAC and SAC augmented with our proposition (SAC-MAMF) in all the domains. In SAC-MAMF, we have currently used the same hyperparameters as in TD3-MAMF. The results of training in the same environments are shown in the attached figures.
>
> In the Mujoco tasks, SAC-MAMF generally improves over SAC, except in Swimmer where the performance of SAC-MAMF is worse. Interestingly, TD3-MAMF is generally better than SAC.
>
> In Dexterous gym, SAC works surprisingly well. In EggCatchUnderarm and EggCatchOverarm, SAC-MAMF generally improves over SAC. InPenSpin, the performances are similar. In all these tasks, the performances of SAC-MAMF have less variance.
>
> We believe that the performance of SAC-MAMF can be further improved and made more stable, by choosing more suitable hyperparameters for our method to adapt it to SAC. We have not done this yet, because running experiments on Dexterous gym is relatively costly: it takes about 4 days on our computers to evaluate one algorithm. And the experiment of running SAC-MAMF in EggCatchOverarm is not finished as shown in the attached figures. We will present those improved results in the final version of our paper.
>
> **To summarize, we have already updated our manuscript to make the presentation clearer (changes are in blue). In the final version, we will update the experimental section to present the improved performance of SAC-MAMF.**
>
>
>
> **Zip File:**
>
> /attachment/faa0d2b314a88355b3d8a480b482d97f9b1b8b3f.zip